# Routing Mamba: Scaling State Space Models with Mixture-of-Experts Projection

**Zheng Zhan**[1,2*†]   **Liliang Ren**[1†]   **Shuohang Wang**[1]   **Liyuan Liu**[1]   **Yang Liu**[1]
**Yeyun Gong**[1]   **Yanzhi Wang**[2]   **Yelong Shen**[1]
[1]Microsoft   [2]Northeastern University

## Abstract

State Space Models (SSMs) offer remarkable performance gains in efficient sequence modeling, with constant per-step inference-time computation and memory complexity. Recent advances, such as Mamba, further enhance SSMs with input-dependent gating and hardware-aware implementations, positioning them as strong alternatives to Transformers for long sequence modeling. However, efficiently scaling the expressive power of SSMs, particularly with Mixture of Experts (MoE), remains challenging, as naive integration attempts often falter or degrade performance. In this work, we introduce Routing Mamba (RoM), a novel approach that scales SSM parameters using sparse mixtures of linear projection experts. By sharing routing decisions between projection layers and lightweight sub-modules within Mamba across experts, RoM leverages synergies among linear projection experts for effective and efficient sparse scaling of Mamba layers. At a scale of 1.3B active parameters (10B total) and 16K training sequence length, RoM achieves language modeling performance equivalent to a dense Mamba model requiring over $2.3\times$ more active parameters, and demonstrates consistent perplexity across context lengths. Experimental results further show RoM effectively scales hybrid language models, yielding a 23% FLOPs saving compared to dense Mamba scaling for similar performance. We release our training codebase at `https://github.com/zhanzheng8585/Routing-Mamba`.

## 1 Introduction

State Space Models (SSMs) have recently gained substantial attention for their efficiency in addressing long-range dependencies and their ability to support parallel training. Originally from the Kalman filter model [24], modern SSMs [17–20, 6] have been successfully applied to diverse sequence modeling tasks across various modalities.

Among these, Mamba [16] has emerged as a notable advancement by introducing time-varying parameters into SSMs, enabling selective memorization of input information. It further integrates a hardware-aware selective scan algorithm that accelerates both training and inference, offering a subquadratic-time architecture that scales efficiently with sequence length. Mamba's scalability allows it to be a competitive alternative to Transformer-based models [44] for generative language modeling in low-resource scenarios.

In parallel to these advancements, Mixture of Experts (MoE) [22, 25, 13] has become a cornerstone for efficiently scaling the number of parameters of large language models [34, 9, 32]. Most contemporary approaches focus on integrating MoE into the feed-forward layers, treating each expert as an independent Feed-forward Network (FFN). Previous works like SwitchHead [5] and Mixture of

---

[*]This work was done during Zheng's internship at Microsoft.

[†]Corresponding authors.

Attention Heads (MoA) [50] extend this paradigm through leveraging attention projections as experts. These works highlight the potential for introducing sparsely activated experts to token-mixing layers for improved performance and efficiency in large language models. Despite these advancements in MoE for Transformers and attention mechanisms, its application to SSMs remains largely unexplored. Specifically, while large-scale SSM models like Mamba [15], Jamba [27], and Samba [39] have demonstrated their efficacy, no prior work has successfully investigated the use of Mamba layers as MoE experts to further scale the number of total model parameters.

To bridge this gap, our study investigates the feasibility of scaling Mamba layers using the sparse mixture of linear projection experts. By leveraging Mamba's intrinsic synergies across sub-modules, we successfully improve model quality and scalability with sparse activation of linear projection layers. This work represents a pioneering effort toward developing large-scale models that utilize Mamba layers as scalable and efficient experts within the sparse MoE framework. We first conduct comprehensive experiments to reveal that applying a straightforward MoE scaling strategy to Mamba layers not only fails to enhance performance but also leads to performance degradation and increased latency overhead, without delivering meaningful improvements. These findings highlight the inherent complexities and challenges of effectively scaling Mamba layers with sparse mixture of projection experts, emphasizing the need for a more tailored approach.

Unlike conventional MoE implementations that primarily focus on feedforward layers, our work systematically studies the sparse scaling of Mamba's unique projection layers, consisting of Convolution (Conv), Gate, and Output projections, and transforms them into sparsely activated experts of linear layers. Through selective scaling of sub-modules and a shared routing strategy, our design significantly improves computational efficiency while expanding model capacity. Beyond addressing the challenges of integrating MoE with SSMs, we also study the effectiveness of RoM in hybrid SSMs-Attention model architectures with various sparse scaling strategies, broadening the horizons for large-scale sequence modeling with SSMs.

With our tailored design, RoM achieves significant improvements in both performance and efficiency. Specifically, across a scale from 115M up to 10B total parameters, RoM demonstrates superior scaling behavior compared to dense Mamba models, achieving equivalent perplexity performance to its dense counterparts that require up to $2.3\times$ more active parameters. RoM also achieves a 23% FLOPs reduction when applied to Samba, a typical Mamba-attention hybrid model. We summarize our contributions as follows:

- We propose the first framework for integrating sparse mixture of projection experts into SSMs, demonstrating their applicability beyond traditional feed-forward networks, shedding light on future research on sparse scaling of SSMs and hybrid models.

- Our selective scaling and shared routing mechanism enable scalable and efficient training for large-scale sequence modeling tasks, achieving model performance that is comparable to up to $2.3\times$ larger dense models.

- We validate the effectiveness of RoM across various configurations, demonstrating its adaptability and generalization. Notably, RoM achieves a 23% reduction in FLOPs for hybrid model scaling, highlighting its potential and setting a new direction for efficient sequence modeling.

## 2 Related Work

**State Space Models.** State Space Models [15, 31, 45] have emerged as a compelling architectural paradigm for sequence-to-sequence transformations. These models are well-suited for capturing complex dynamics in sequential data. Mamba-2 [6] extends this foundation by leveraging state space duality, introducing a refined architecture based on selective SSMs that demonstrates both versatility and scalability. Recently, SSM-based Small Language Models (SLM) such as Gated DeltaNet [47], Jamba [27], Samba [39], and Hymba [10] have shown significant scalability and effectiveness. These models adhere to scaling laws and have proven particularly promising for small-scale language modeling, offering a balance of performance and cost-efficiency for diverse and complex tasks.

Despite these advancements, previous works [27, 37] fail to successfully integrate MoE with the SSMs. Considering that SSM is the fundamental component in these architectures, incorporating MoE with SSMs presents an intuitive and innovative way to enhance performance while maintaining

computational efficiency. This integration has the potential to combine the strengths of both paradigms, enabling more scalable and resource-efficient solutions.

**Mixture of Expert.** Mixture of Experts models have emerged as a powerful technique in LLMs. The original concept of MoE [22] involves partitioning the input space into regions and training specialized experts for each region, with a gating network to dynamically select the most suitable expert for a given input. Recent advancements [42, 23, 11, 13] have leveraged the MoE framework to address the increasing complexity and scale of modern datasets. Extensions like GShard [25] have adapted MoE to multilingual settings, enabling them to scale effectively across over 100 languages and achieve SOTA results in multilingual benchmarks. Efforts such as SwitchHead [5] and Mixture of Attention Heads (MoA) [50] aim to harness sparse computation to improve attention capacity. However, their scalability remains limited, leaving room for further exploration of the expert architectures.

Existing SSM models [27, 37, 1] including MoE focused on combining the FFN layer and the MoE. MoM [26] is a recent work that employs a rule-based sparsity approach to integrate expert and Mamba for each modality. However, it lacks scalability and generalization ability due to the rule-based strategy. Our work introduces the RoM, addressing a critical gap in the literature. It aims to expand the capabilities of SSMs, transforming them into highly scalable and efficient components for large-scale sequence modeling.

## 3 Preliminaries

### 3.1 Mamba Layer

Mamba [15] is a recently proposed SSM-based model with selective state spaces. It represents a discrete version of the continuous system for SSMs and incorporates a timescale parameter $\Delta$ to facilitate the transformation of continuous parameters with the zero-order hold. The layer types within Mamba are shown as follows:

- **Conv Proj** ($\mathbf{W}_{in}$) and **Gate Proj** ($\mathbf{W}_g$): These layers are coupled together to form a single large layer.
- **Out Proj** ($\mathbf{W}_{out}$): Responsible for the output projections.
- **x Proj, dt Proj, Conv 1D**: Additional specialized layers within the Mamba layer and SSM selective scan.

Given an input sequence representation $\mathbf{X} \in \mathbb{R}^{L \times D_m}$, where $L$ is the length of the sequence and $D_m$ is the hidden size, Mamba first expands the inputs to a higher dimension $D_e$, i.e.,

$$\mathbf{H} = \mathbf{X}\mathbf{W}_{in}, \tag{1}$$

where $\mathbf{W}_{in} \in \mathbb{R}^{D_m \times D_e}$ is a learnable projection matrix. Then a Short Convolution (SC) [38] operator is applied to smooth the input signal,

$$\mathbf{U} = \text{SC}(\mathbf{H}) = \text{SiLU}(\text{DWConv}(\mathbf{H}, \mathbf{W}_{conv})), \tag{2}$$

where $\mathbf{W}_{conv} \in \mathbb{R}^{k \times D_e}$ and the kernel size $k$ are set to 4 for hardware-aware efficiency. The Depthwise Convolution [3] is applied over the sequence dimension followed by a SiLU [12] activation function. And then $\mathbf{U}$ goes through SSMs for data-dependent context modeling. It processes $\mathbf{U}$ from the `forward` scan via:

$$\mathbf{Y} = \text{SSM}(\mathbf{A}, \mathbf{B}, \mathbf{C})(\mathbf{U}), \tag{3}$$

where the hidden states $\mathbf{Y}$ are the output of SSM. SSM maps an input sequence $u(t) \in \mathbb{R}^L$ to an output sequence $y(t) \in \mathbb{R}^L$ through a hidden state $h(t)$ as follows,

$$h'(t) = \mathbf{A}h(t) + \mathbf{B}u(t), \quad y(t) = \mathbf{C}h(t), \tag{4}$$

where $L$ denotes the length of the sequence, $\mathbf{A} \in \mathbb{R}^{D_e \times D_s}$ is a learnable matrix, and $\mathbf{B} \in \mathbb{R}^{L \times D_s}$, $\mathbf{C} \in \mathbb{R}^{L \times D_s}$ are the SSM parameters. Mamba incorporates a timescale parameter $\Delta$ to facilitate the transformation of continuous parameters with the zero-order hold (ZOH) as $\overline{\mathbf{A}} = \exp(\Delta\mathbf{A})$, and $\overline{\mathbf{B}} = (\Delta\mathbf{A})^{-1}(\exp(\Delta\mathbf{A}) - \mathbf{I}) \cdot \Delta\mathbf{B}$. After the parameters have been transformed, the model can be computed in a linear recurrence.

**Linear recurrence**: the discretization of Equation (4) can be rewritten as,

$$\mathbf{h}_t = \overline{\mathbf{A}}\mathbf{h}_{t-1} + \overline{\mathbf{B}}\mathbf{U}_t, \quad \mathbf{Y}_t = \mathbf{C}\mathbf{h}_t. \tag{5}$$

In practice, Mamba implements a hardware-aware parallel scan algorithm for efficient parallelizable training. The final output is obtained through a gating mechanism similar to Gated Linear Unit [41, 7],

$$\mathbf{O} = \mathbf{Y} \odot \text{SiLU}(\mathbf{X}\mathbf{W}_g)\mathbf{W}_{\text{out}}, \tag{6}$$

where $\mathbf{W}_g \in \mathbb{R}^{D_m \times D_e}$ and $\mathbf{W}_{\text{out}} \in \mathbb{R}^{D_e \times D_m}$ are learnable parameters. Here we set $d_e = 2d_m$, $d_r = d_m/16$, and $d_s = 16$. The Mamba layer leverages its recurrent structure to capture the temporal semantics of the input sequence. Its input selection mechanism enables the model to focus on relevant inputs, enhancing its ability to retain information over extended time spans.

## 3.2 Mixture of Experts

MoE is proposed to scale up the model capacity while maintaining its cost-effectiveness. A traditional MoE block consists of $n$ experts $\{E_1, E_2, ..., E_n\}$, each of which is a FFN similar to those in the transformer block. Giving an input embedding $x$, it is fed into a router network $\mathcal{G}$ and assigned to the most relevant experts. The architecture of the router network is usually one or a few layers of multi-layer perceptrons (MLPs). The gating mechanism is defined as follows:

$$\mathcal{R}(\boldsymbol{x}) = \text{TopK}(\text{Softmax}(g(\boldsymbol{x}))) \tag{7}$$

where $g(\cdot)$ are trainable gating networks and TopK select the largest $K$ values. The final output of an MoE is the weighted sum of features from the activated experts and can be depicted as below:

$$\boldsymbol{y} = \sum_{i=1}^{N} \mathcal{R}_i(\boldsymbol{x}) \cdot E_i(\boldsymbol{x}) \tag{8}$$

where $E_i(\boldsymbol{x})$ stands for the feature representations produced from the expert, which is weighted by $\mathcal{R}(\boldsymbol{x})$ to form the final output $\boldsymbol{y}$.

# 4 Design of Routing Mamba

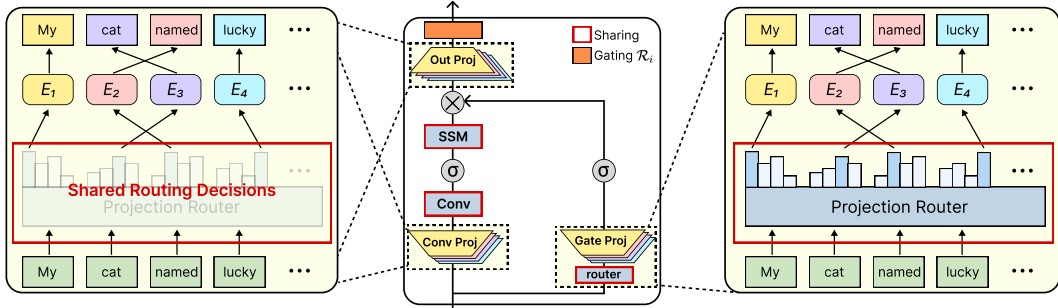

Figure 1: Our method overview. The red box indicates components that are shared across all experts. (*Left*) This section illustrates how our method applies the shared routing strategy to the **Conv Proj** and **Out Proj** layers. (*Middle*) This part presents the Mamba block incorporated in our method, where we apply the gating weights to multiply with the representations after **Out Proj**. (*Right*) This section demonstrates the **Gate Proj** under our method. For instance, the token "cat" is first processed by the projection router, which then determines that it should be routed to the third expert.

The overview of Routing Mamba is illustrated in Figure 1, which emphasizes the key projection layers and their distinct functions. Unlike FFN layers, which consist of a single type of operation, a Mamba layer integrates multiple types of projection layers, each serving a unique function within the SSM forward process. This architectural complexity poses both opportunities and challenges when scaling Mamba layers using MoE.

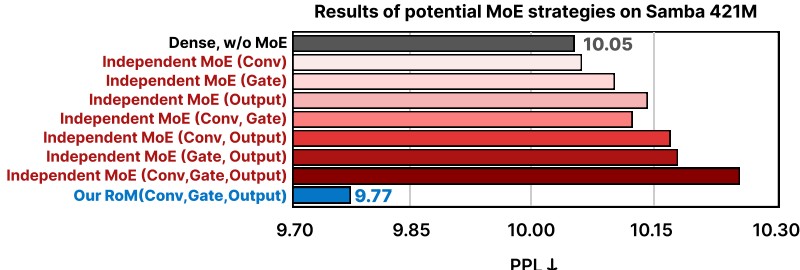

Figure 2: Perplexity (PPL) on the SlimPajama validation set (with 20B tokens pretrained) with different MoE strategies. Bars in red depict the performance when applying MoE independently to specific Mamba projection layers (e.g., Conv, Gate, or combinations, as labeled), same as the MoE-Mamba approach [37]. Experimental setup and detailed definitions of strategies are provided in Section 4.1. Further details on these comparisons are in **Appendix A.1**.

## 4.1 The Challenge of Naive MoE Integration in Mamba

While Mixture of Experts effectively scales Transformers, its direct application to SSMs like Mamba is notably challenging. Figure 2 illustrates this by comparing perplexity (PPL) on the SlimPajama validation set (20B tokens) for the Samba 421M model under several configurations. These include a Dense baseline which represents the Samba 421M model without any MoE layers and strategies depicted by red bars where MoE is applied independently to Mamba's Conv, Gate, Output projection layers, or their combinations, mirroring the MoE-Mamba approach [37].

Consistent with prior studies [27, 37] and as evidenced in Figure 2, **such naive, independent application of MoE to Mamba's projection layers generally fails to improve performance, often degrading it and potentially increasing latency.** This difficulty underscores that Mamba's architectural intricacies—particularly the complex interplay and specialized roles of its projection layers—resist straightforward MoE integration. **Effectively scaling Mamba with MoE is therefore a non-trivial task** that necessitates a tailored design, motivating the development of our RoM approach.

## 4.2 RoM Methodology

Our approach tackles the challenge of efficiently scaling Mamba layers with RoM by selectively targeting components that offer the highest potential for performance gains. Instead of uniformly scaling all parameters in the Mamba layer, we focus on specific layers with the greatest computational and representational impact, enabling a more strategic and effective utilization of a sparse mixture of linear projection experts. Furthermore, our method incorporates a shared routing strategy. By reusing routing decisions across projection layers, we achieve a dual benefit: lowering the difficulty of router learning while maintaining high-quality expert allocation.

**RoM routing mechanism.** In the RoM routing mechanism, only the top $K$ experts with the highest gating weights $\mathcal{R}_i(\mathbf{X}_t)$ are selected. To implement this, the routing strategy normalizes the gating weights over the selected $K$ experts, while setting the weights of the inactive experts to zero. In brief, the router assigns a subset of experts to each input based on their relevance, computed dynamically for every time step $t$. The routing mechanism is defined as:

$$\mathcal{R}_i(\mathbf{X}_t) = \mathcal{P}(\mathbf{X}_t) \cdot \mathbf{1}_{i \in \text{TopK}(\mathcal{P})}, \tag{9}$$

where $\mathcal{P}(\mathbf{X}_t) = \text{Softmax}(\mathbf{X}_t \cdot \mathbf{W}_r)$ represents the softmax probabilities, which can be translated as the relative importance of each expert. These weights are computed by projecting the input $\mathbf{X}_t$ through a set of learnable parameters $\mathbf{W}_r \in \mathbb{R}^{D_m \times N}$ associated with each expert. And $\mathbf{1}_{i \in \text{TopK}(\mathcal{P})}$ is an indicator function that is 1 if $i$ belongs to the top $K$ experts and 0 otherwise. This indicator will be used later in the shared routing strategy.

With the selective scaling and shared routing decisions, we can derive the RoM formulation as follows. First, $\mathbf{G}$ represents the outputs of the **Gate Proj**, computed as:

$$\mathbf{G} = \text{SiLU}(\sum_{i=1}^{N} \mathbf{1}_{i \in \text{TopK}(\text{Softmax}(\mathbf{X}_t \cdot \mathbf{W}_r))} \cdot \mathbf{X}_t \mathbf{W}_{g,i}). \tag{10}$$

Where $\mathbf{W}_r$ are the learnable weights of the router, we adopt the shared routing decisions for the later projection layer with $\mathbf{1}_{i \in \text{TopK}(\mathcal{P}_{\mathbf{G}})} = \mathbf{1}_{i \in \text{TopK}(\text{Softmax}(\mathbf{X}_t \cdot \mathbf{W}_r))}$ from **Gate Proj**. Additionally, the intermediate hidden $\mathbf{H}_t$ representation after the **Conv Proj** is defined as:

$$\mathbf{H}_t = \sum_{i=1}^{N} \mathbf{1}_{i \in \text{TopK}(\mathcal{P}_{\mathbf{G}})} \cdot \mathbf{X}_t \mathbf{W}_{\text{in},i}. \tag{11}$$

We adopt the shared routing decisions in the **Conv Proj** layer. The output of the RoM is the weighted sum of the selected experts' outputs. Formally, the RoM output at time step $t$ can be written as:

$$\mathbf{O}_t = \sum_{i=1}^{N} \mathcal{R}_i(\mathbf{X}_t) \cdot E_i(\mathbf{Y}_t, \mathbf{X}_t) \tag{12}$$

where $\mathcal{R}_i(\mathbf{X}_t)$ represents the gating weight for the $i$-th expert in the **Gate Proj**. We adopt the shared routing decisions in the **Out Proj** layer again, but this time we gate the results from selected experts. And the expert computation $E_i(\mathbf{Y}_t, \mathbf{X}_t)$ is defined as:

$$E_i(\mathbf{Y}_t, \mathbf{X}_t) = \mathbf{Y}_t \odot \mathbf{G} \mathbf{W}_{\text{out},i} \tag{13}$$

By replacing Eq. 1 with Eq. 11 and Eq. 6 with Eq. 10 12 13, we can get a basic and effective RoM strategy. This formulation underscores the importance of leveraging the sparsity of the mixture of linear projection experts while carefully balancing the computational trade-offs. Moreover, this leaves space for the future development of the Mamba architecture and hybrid model scaling for Mamba layers. By prioritizing impactful components and optimizing the routing mechanism, our approach advances the scalability and efficiency of SSMs.

### 4.3 Details of Design Choices

**RoM design choices.** For the intermediate computations involving **x Proj**, **dt Proj**, and **Conv 1D**, these layers have relatively few parameters. Inspired by the Multi-Query Attention mechanism, we propose sharing a single set of parameters across all experts for these components. This mirrors the concept of sharing key and value heads across multiple query heads in Multi-Query Attention. We also provide studies on the effect of these parts in Table 1. More router training details can be found in **Appendix A.3**.

**Load balance.** Previous work [42] has observed that the routing network tends to converge to a state where it always produces large weights for the same few experts, which indicates the insufficient utility of all the experts. However, in this work, we feature RoM without any load balance loss. We also provide results with the load balance loss (auxiliary loss) in **Appendix A.3** to demonstrate the inherent ability of RoM to naturally balance the load across different experts.

## 5 Experiments

### 5.1 Implementation Details

We choose PyTorch Fully Sharded Data Parallel (FSDP) with CPU offloading as a scalable framework for all the model training. We avoid expert parallelism to eliminate the need for a capacity factor or token dropping, ensuring more stable training and improved performance across all configurations and baselines. For MoE computation without expert parallelism, we find the Megablocks [14] package to be helpful. Particularly, we leverage its `grouped_GEMM` kernels which offer acceleration improvement.

**Models and datasets.** We conduct experiments across models of varying scales, encompassing different parameter sizes and architectures. Details can be found in **Appendix A.2**. Following the settings outlined in [39], we report perplexity results on the SlimPajama [43] dataset, observing minimal fluctuation across evaluations. Training speed is measured using $8 \times$A100 GPUs. By default, the models in this section are trained on SlimPajama using 20B tokens and a sequence length of 4K, unless stated otherwise.

**Training details.** Our model is optimized using the AdamW optimizer with beta1 = 0.9 and beta2 = 0.95, with gradient clipping set to 1.0 and weight decay to 0.1. A cosine learning rate schedule is

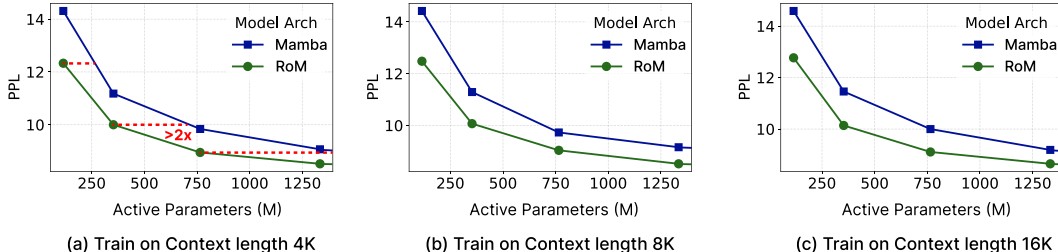

(a) Train on Context length 4K    (b) Train on Context length 8K    (c) Train on Context length 16K

Figure 3: Validation Perplexity (PPL) of various scales of RoM and Mamba models with different training sequence length (4K, 8K, 16K) pre-trained on 20B tokens of SlimPajama. Our method shows consistency over all training sequence length. RoM has significantly better PPL over the Mamba and the Mamba model requires at least $2\times$ active parameters to achieve the similar results compared with RoM model. For RoM models, we activates 1 out of 8 expert networks for each token at each layer.

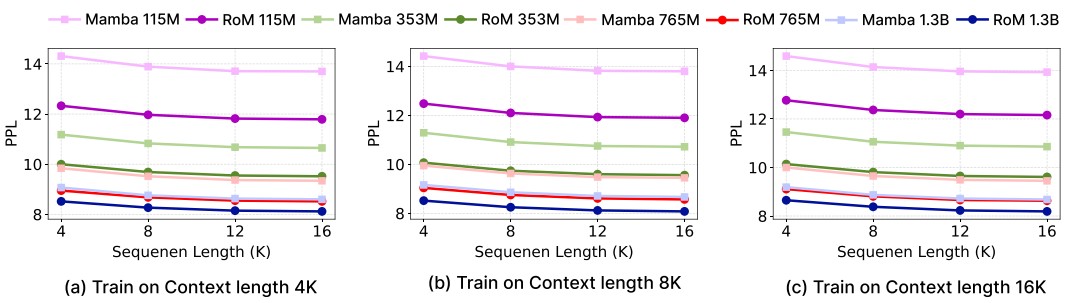

(a) Train on Context length 4K    (b) Train on Context length 8K    (c) Train on Context length 16K

Figure 4: Validation perplexity of various evaluation sequence lengths for models pre-trained with different sequence length (4K, 8K, 16K). Our method shows consistency over all three different training settings. For RoM model, we activates 1 out of 8 expert networks for each token at each layer. Detailed results can be found in **Appendix A.4**

employed, with a maximum learning rate of 4e-4 for models and a warmup ratio of 0.01. We train our models with a global batch size of 2 million tokens, unless stated otherwise.

**Evaluation benchmarks.** We evaluate task performance on several common sense reasoning datasets, including LAMBADA [35], HellaSwag [48], PIQA [2], ARC-Easy [4], ARC-Challenge [4], and WinoGrande [40]. We report perplexity on the LAMBADA dataset and average accuracy across all the mentioned datasets.

## 5.2 Scaling on Mamba

We also conduct experiments for RoM scaling on Mamba where directly applied traditional MoE is not feasible since there is no FFN layer. We include four configurations with varying parameter sizes: 115M, 353M, 765M, and 1.3B. Each configuration is defined by the number of layers {24, 48, 48, 48}, hidden size {768, 1024, 1536, 2048} with d_state = 16. For instance, the 115M model has 12 layers, a hidden size of 768. For RoM model, we activate 1 out of 8 expert networks for each token at each layer. From Figure 3 and Figure 4, we can observe that RoM can greatly improve the performance over the Mamba model.

Figure 3 presents the PPL results on the validation dataset, comparing RoM against a standard Mamba model across different training sequence lengths (4K, 8K, and 16K). The results clearly show that RoM consistently outperforms the Mamba models across all active parameter scales. Notably, the red dashed line highlights that the Mamba model requires at least $2\times$ **more active parameters** to reach a similar perplexity as RoM, showcasing the efficiency gains provided by RoM-based scaling. Specifically, for Figure 3(a), we achieve an equivalent perplexity performance as its dense counterpart with up to $2.3\times$ active parameters for 115M scale. For 353M scale, we achieve $2.0\times$, respectively. This efficiency advantage persists across all training contexts, demonstrating RoM 's capability to effectively scale state-space models while maintaining competitive model quality.

Beyond training efficiency, Figure 4 evaluates the length extrapolation ability of RoM, testing how well models generalize to longer sequence lengths than those seen during training. RoM exhibits

Table 1: Comparison of FLOPs (one forward pass with seq_length = 4K) and validation context lengths for various architectures trained on the SlimPajama dataset.

| Architecture | Active Params. | Total Params. | FLOPs | Validation Context Length | | |
|---|---|---|---|---|---|---|
| | | | | 4096 | 8192 | 16384 |
| **20B training tokens on 8×A100 GPUs** | | | | | | |
| Llama-2 | 438M | 438M | 4.42T | 11.14 | 47.23 | 249.03 |
| Mamba | 432M | 432M | 6.49T | 10.70 | 10.30 | 10.24 |
| Samba (expand=2) | 421M | 421M | 4.74T | 10.05 | 9.64 | 9.56 |
| + MoA | 421M | 1.1B | 4.74T | 9.94 | 9.54 | 9.46 |
| + SwitchHead | 421M | 1.1B | 4.74T | 9.84 | 9.45 | 9.37 |
| + MoE-Mamba (Conv, Gate, Out) | 421M | 1.0B | 4.74T | 10.26 | 9.85 | 9.77 |
| + RoM (Conv, Gate, Out) | 421M | 1.0B | 4.74T | **9.77** | **9.38** | **9.31** |
| Samba (expand=4) | 511M | 511M | 6.13T | 9.78 | 9.39 | 9.31 |
| + RoM (Gate, Out) | 511M | 1.3B | 6.13T | 9.37 | 9.00 | 8.93 |
| + RoM (Conv, Gate, Out) | 511M | 1.7B | 6.13T | 9.28 | 8.92 | 8.85 |
| + RoM (Conv, Gate, dt, x, Out) | 511M | 1.7B | 6.13T | 9.30 | 8.93 | 8.87 |

superior generalization across all model sizes, maintaining lower perplexity compared to the Mamba model at longer sequence lengths.

These results reinforce several key points. First, RoM achieves lower perplexity while requiring significantly fewer active parameters, making it a more efficient alternative to traditional Mamba architectures. Second, RoM scales effectively across different training sequence lengths, ensuring consistent performance improvements. Lastly, RoM 's ability to extrapolate to longer sequences highlights the benefits of our method. This combination of efficiency, scalability, and robustness positions RoM as a powerful framework for scalable and resource-efficient long-sequence modeling.

## 5.3 Comparison and Ablation Study

Table 2: Evaluation of downstream task performance for FFN-MoE and hybrid RoM + MoE architectures across different sparsity level of experts on Samba 511M.

| Method | Active Params. | Total Params. | LAMBADA | | HellaSwag | PIQA | Arc-E | WinoGrade | Avg. |
|---|---|---|---|---|---|---|---|---|---|
| | | | PPL↓ | Acc↑(%) | Acc↑(%) | Acc↑(%) | Acc↑(%) | Acc↑(%) | Acc↑(%) |
| FFN-MoE (16top1) | 511M | 2.8B | 24.8 | 38.4 | 42.1 | 68.1 | 50.1 | 52.3 | 50.2 |
| RoM + FFN-MoE (8top1) | 511M | 2.9B | 24.8 | 39.5 | 41.6 | 68.1 | 48.7 | 52.8 | 50.1 |
| FFN-MoE (32top1) | 511M | 5.7B | 26.9 | 38.0 | 40.7 | 67.4 | 50.3 | 51.1 | 49.5 |
| RoM + FFN-MoE (16top1) | 511M | 5.6B | 28.5 | 36.4 | 41.0 | 68.0 | 48.4 | 52.6 | 49.2 |

In Table 1, we compare various architectures—including traditional Mamba, hybrid models, and attention-based MoE variants like MoA and SwitchHead. Architectures overview can be found in **Appendix A.2**. We report active/total parameters, FLOPs (for sequence length 4K), and validation accuracy at context lengths 4096, 8192, and 16384. Active parameters are those used during inference; while total parameters include all components of the architecture. The experiments evaluate the impact of integrating RoM into different Mamba projection layers, namely Conv, Gate, and Out projections.

Figures 3 and 4 show that RoM integrates well with Mamba and **extends seamlessly to hybrid models like Samba**, unlike MoE-Mamba [37]. It not only surpasses Llama-2 and Mamba but also improves Samba with high cost-efficiency. This highlights RoM's ability to scale efficiently while maintaining superior performance, reinforcing its potential as a robust and versatile approach for improving hybrid language models. We also adapt MoA and SwitchHead for Samba's attention layers, aligning total parameter counts (e.g., Top-1 over 32 experts) to ensure a fair comparison. For RoM, we use Top-1 over 8 to balance efficiency and expressiveness. Notably, RoM consistently outperforms MoA and SwitchHead across all validation lengths.

**RoM Scaling Outperforms Mamba Layer Expanding.** A key insight is that RoM not only surpasses MoA and SwitchHead but also outperforms simple Mamba layer expanding. For instance, RoM (Conv, Gate, Out) achieves comparable PPL to Samba (expand=4), and saves the FLOPs by 23%. This highlights a crucial advantage of RoM: it enables efficient scaling by selectively applying MoE only where it yields the higher gains, rather than indiscriminately increasing model size. By

optimizing sparse mixture of expert allocation, RoM enhances model quality while maintaining a significantly lower computation cost, demonstrating a more scalable and cost-effective approach to SSM scaling. The Table 10 and Table 2 provide a detailed comparison of model performance and training efficiency across different Hybrid MoE configurations, focusing on the combination of RoM with feed-forward MoE (FFN-MoE). We carefully align the total parameter count of FFN-MoE and hybrid RoM + FFN-MoE by adjusting the expert number and keeping the last few layers dense.

**Hybrid RoM + FFN-MoE for Downstream Task Performance.** Table 2 further explores the effectiveness of RoM in downstream tasks, including LAMBADA, HellaSwag, PIQA, Arc-E, and WinoGrade. The Samba + RoM + FFN-MoE (8top1) configuration achieves an average accuracy of 50.1%, closely matching the FFN-MoE (16top1) baseline (50.2%). More notably, in the larger-scale setup (5.6B vs. 5.7B total parameters), Hybrid RoM + FFN-MoE (16top1) reaches an average accuracy of 49.2%, maintaining the same level of performance as FFN-MoE (32top1) at 49.5% in downstream tasks.

**Why choose hybrid models such as Samba as a baseline?** The development of hybrid architectures integrating SSMs with attention mechanisms is a rapidly advancing frontier in sequence modeling [30, 27, 10, 36, 33, 8]. These efforts aim to combine the complementary strengths of SSMs and attention to create more scalable, generalizable, and powerful language models. Samba [39], which interleaves Mamba with Sliding Window Attention (SWA) and MLPs, is a notable example of this trend. Its strong performance with linear computation complexity demonstrates the potential of hybrid model architecture. This makes it a good baseline for applying RoM to study how our approach can affect the length extrapolation performance of linear complexity models with increased representation power of the recurrent states.

**Training Throughput and Overhead Analysis.** As summarized in Table 11, we compare RoM model to their corresponding dense models with the same number of active parameters, measuring their training throughput using the identical hardware. Despite having over two times as many total parameters as the dense model, RoM model achieves nearly 80% relative throughput in this experiment without optimization, confirming the significant computational scaling potential of models with the RoM method.

**RoM on other linear recurrent architectures.** Table 3 shows that RoM significantly boosts the performance of various SSM-based architectures by efficiently scaling them with MoE, demonstrating its effectiveness and broad applicability on the models with the Mamba style of parameterization.

Table 3: Results of performance for SSM-based architectures and RoM.

| Architecture | Active Params. | Total Params. | Validation Context Length | | | |
|---|---|---|---|---|---|---|
| | | | 4096 | 8192 | 12288 | 16384 |
| Mamba | 353M | 353M | 11.18 | 10.83 | 10.68 | 10.65 |
| Mamba + RoM | 353M | 2.5B | 10.00 | 9.69 | 9.55 | 9.52 |
| Mamba2 + RoM | 352M | 2.5B | 9.82 | 9.49 | 9.34 | 9.30 |
| Gated DeltaNet + RoM | 343M | 2.5B | 9.81 | 9.46 | 9.29 | 9.26 |

**Why RoM works better than previous methods?** The efficacy of RoM, particularly its shared routing mechanism across the Conv, Gate, and Output projection layers, can be partly understood by an analogy to MoE in MLPs. In a hypothetical scenario where Mamba operates purely token-wise for these projections, without sequential dependencies, this set of operations resembles parallel MLPs applied to each token. Standard MoE implementations for MLP blocks in Transformers employ a single router that assigns each token to a complete expert MLP, a strategy proven effective for learning specialized token representations.

In contrast, prior approaches to MoE in Mamba, or naive integrations, might involve independent routers for each Mamba projection layer (e.g., separate routers for Conv, Gate, and Output expert sets). Such an independent routing scheme significantly increases the learning burden on each router. More critically, it can hinder the development of coherent token-wise specialization. The Mamba projection layers are functionally interdependent; independent routing decisions could lead to a token's representation being processed by conflicting or uncoordinated expert pathways across these layers. This makes it difficult for the overall Mamba block to learn consistent, specialized expert functions for different types of tokens, effectively fragmenting the routing logic.

RoM's shared routing strategy circumvents these issues. By employing a single router to select a unified "expert pathway" encompassing the specialized Conv, Gate, and Output projections, RoM aligns the MoE application with the successful paradigm of holistic expert selection for a composite operation. This simplifies the router's task to identifying the appropriate expert group for each token, fostering synergistic co-adaptation among the expert projections. Consequently, these expert pathways can specialize more effectively and coherently, leading to improved performance and training stability, as observed in our experiments.

### 5.4 Empirical Insights on Applying RoM-style MoE Across Architectures

Our investigation into applying RoM's shared routing strategy for MoE yields several key empirical insights that vary with architectural choices, offering guidance for future model design:

**Comprehensive Expertization for Streamlined SSMs.** In Mamba-like architectures that feature a more unified block design, such as Mamba2 and Gated DeltaNet, we observe optimal performance when RoM is applied comprehensively across all major projection layers within the block. As indicated in Table 3, converting these primary computational pathways into experts under a shared routing mechanism allows these models to effectively leverage expert capacity. This suggests that for SSMs with tightly integrated projection paths, a holistic MoE application via RoM is beneficial.

**Selective Expertization for Original Mamba.** For the original Mamba architecture , which includes smaller, independently parameterized projections like **x Proj** and **dt Proj** alongside the main Conv, Gate, and Output layers , our RoM strategy focuses on expertizing only the latter, larger projections while sharing the parameters of **x Proj**, **dt Proj**, and the 1D convolution across all experts (Section 4.3). Our ablations (e.g., Table 1, "+ RoM (Conv, Gate, dt, x, Out)" ) show that including these smaller, specialized parameter sets as part of the MoE experts, rather than sharing them, can be slightly detrimental. **This finding could shed light on the future architecture design for MoE-friendly SSMs.**

**Holistic MoE for Self-Attention and MLP Blocks.** When considering the application of MoE to self-attention mechanisms or standard MLP blocks (e.g., those in Transformer FFN layers), our experiments suggest that a holistic approach to expertization is generally more effective than partial or piecemeal MoE integration within the block. For instance, attempting to apply MoE to only a subset of layers within a self-attention block (e.g., expertizing two of four projection matrices while sharing the others) does not yield clear performance improvements. Similarly, for MLP blocks, applying MoE comprehensively to all constituent large projection layers (e.g., both the up- and down-projection matrices in an FFN) under a shared router aligns with established best practices and performs well. This indicates that for these types of computational blocks, the entire functional unit benefits from being treated as a single entity for expert-based scaling.

Collectively, these observations suggest that the optimal strategy for integrating MoE with shared routing, as pioneered by RoM, is context-dependent. Architectures with large, computationally significant, and functionally cohesive projection pathways (like those in Mamba2, Gated DeltaNet, or entire MLP blocks) benefit from comprehensive expertization. In contrast, for architectures like the original Mamba with very small, specialized parametric components, a selective approach that expertizes major projections while sharing minor ones appears more advantageous. Furthermore, partial or uncoordinated MoE application within tightly coupled multi-layer operations, such as self-attention, may not be effective.

## 6 Conclusion

We introduce Routing Mamba (RoM), a novel framework that integrates MoE mechanisms into SSMs by leveraging Mamba's projection layers as scalable expert components. Through selective scaling and a shared routing strategy, RoM addresses the ineffectiveness and performance degradation of naive MoE integration, achieving substantial gains in efficiency and performance on various benchmarks. It also demonstrates consistent length extrapolation ability across scales. By bridging both paradigms, RoM establishes a flexible framework that sets a new direction for cost-effective sparse scaling of large language modeling with SSMs. Limitations remain due to uncertainty around RoM's optimal configuration and its broader applicability to the rapidly evolving landscape of SSM variants, as well as broader self-attention and linear attention architectures.

# 7 Acknowledgments

We thank Ziyi Yang, Songlin Yang, and Chen Liang for their helpful feedback.

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

# A Appendix

## A.1 Detailed Comparisons Results for Potential RoM Strategies

Table 4: Comparison of validation context lengths for various potential RoM architectures trained on the SlimPajama dataset with Samba 421M. For the MoE-Mamba and RoM model, we activate only 1 out of 8 expert networks per token at each layer.

| Architecture | Active Params. | Total Params. | Validation Context Length | | |
| --- | --- | --- | --- | --- | --- |
| | | | 4096 | 8192 | 16384 |
| **20B training tokens on 8×A100 GPUs** | | | | | |
| Samba (expand=2) | 421M | 421M | 10.05 | 9.64 | 9.56 |
| + MoE-Mamba (Conv) | 421M | 620M | 10.06 | 9.66 | 9.58 |
| + MoE-Mamba (Gate) | 421M | 620M | 10.10 | 9.69 | 9.60 |
| + MoE-Mamba (Out) | 421M | 620M | 10.14 | 9.73 | 9.65 |
| + MoE-Mamba (Conv, Gate) | 421M | 818M | 10.12 | 9.70 | 9.62 |
| + MoE-Mamba (Conv, Out) | 421M | 818M | 10.17 | 9.75 | 9.68 |
| + MoE-Mamba (Gate, Out) | 421M | 818M | 10.18 | 9.77 | 9.69 |
| + MoE-Mamba (Conv, Gate, Out) | 421M | 1.0B | 10.26 | 9.85 | 9.77 |
| + RoM (Conv, Gate, Out) | 421M | 1.0B | **9.77** | **9.38** | **9.31** |

Table 4 presents a comparative analysis of different RoM strategies applied to the Samba model with 421M active parameters. The experiments evaluate the impact of integrating MoE mechanisms into different Mamba projection layers, namely **Conv** (Conv Projection), **Gate** (Gate Projection), and **Out** (Output Projection). The implementation and results can be validated in MoE-Mamba [37] Table 3.

**Effect of Different Components.** The introduction of MoE mechanisms to individual projection components provides marginal yet notable degradations over the baseline Samba model. When multiple MoE components are combined, the degradations become more pronounced.

**Comparison of RoM and MoE-Mamba Strategies.** The RoM (Conv, Gate, Out) strategy delivers superior validation scores across all tested context lengths. Notably, at 16K length, RoM (Conv, Gate, Out) achieves a validation score of 9.31, outperforming MoE-Mamba (Conv, Gate, Out) at 9.77. Despite both approaches operating with the same total parameter count (1.0B). It is obvious that a straightforward MoE scaling strategy for Mamba layers fails to enhance performance but also degrades it. This highlights the challenge of effectively scaling Mamba layers, as their complex interplay of projection types necessitates careful optimization. The inherent complexity of Mamba layers makes integrating MoE non-trivial, requiring a well-designed approach that strategically optimizes interactions between projections.

## A.2 Architectures Overview

As illustrated in Figure 5, the first row (from left to right) consists of Mamba, Samba, Samba+FFN-MoE, Samba+MoA, and Samba+SwitchHead, while the second row includes RoM, Samba+RoM, and Samba+RoM+FFN-MoE. These diagrams showcase the layer-wise integration of Mamba with different configurations of FFN and Sliding Window Attention (SWA). For clarity, embedding layers and output projections are omitted. Additionally, Pre-Norm [46, 49] and skip connections [21] are applied to each intermediate layer. Deploying RoM on Mamba2 or Gated DeltaNet [47] follows the same logic as deploying it on the original Mamba, due to their similar architectural designs.

To be noted, in Samba+RoM+FFN-MoE, we adopt the shared routing decisions strategy from RoM to FFN-MoE for better convergence. The formulation can be defined as follows at time step $t$:

$$\mathbf{Y}_t = \sum_{i=1}^{N} \mathcal{R}_i(\mathbf{X}_t) \cdot E_i(\mathbf{X}_t), \tag{14}$$

where $\mathbf{Y}_t$ is the output of feedforward SwiGLU MoE. And the expert computation is defined as:

$$E_i(\mathbf{X}_t) = \sum_{i=1}^{N} \mathbf{1}_{i \in \text{TopK}(\mathcal{P}_\mathbf{G})} \cdot f_i(\mathbf{X}_t), \tag{15}$$

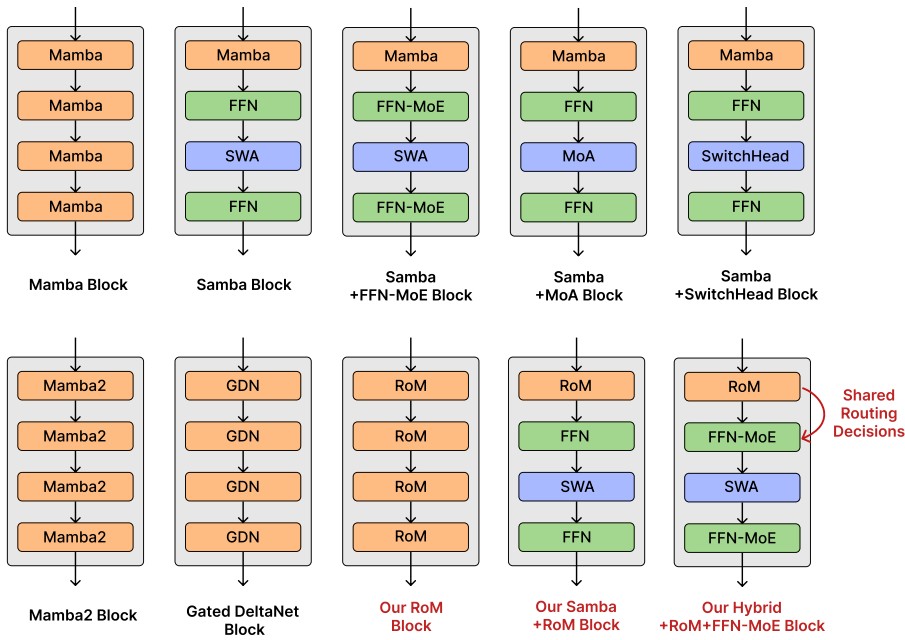

Figure 5: Architectures overview of all configurations and baseline models.

where the $f_i$ is $i$-th feedforward SwiGLU expert and $\mathbf{1}_{i \in \text{TopK}(\mathcal{P}_{\mathbf{G}})} = \mathbf{1}_{i \in \text{TopK}(\text{Softmax}(\mathbf{X}_t \cdot \mathbf{W}_r))}$ is the shared routing decisions strategy from the Gate projection layer in the previous RoM layer.

Table 5: (Scaling Law Model Sizes for Mamba.) Our Mamba model sizes and hyperparameters for scaling experiments.

| Params | n_layers | d_model |
|--------|----------|---------|
| 115M | 24 | 768 |
| 353M | 48 | 1024 |
| 765M | 48 | 1536 |
| 1.3B | 48 | 2048 |

RoM configurations follow this scaling configurations accordingly. Additional details are provided here for all configurations, including training steps (9535), learning rate (4e-4), batch size (2 million tokens), and a total of 20 billion tokens used for training.

## A.3  Load Balance and More Router Details

Table 6: Comparison of validation context lengths with or without load balance loss.

| Architecture | Active Params. | Total Params. | Validation Context Length | | |
|--------------|----------------|---------------|------|------|-------|
| | | | 4096 | 8192 | 16384 |
| Samba (expand=4) | 511M | 511M | 9.78 | 9.39 | 9.31 |
| + RoM (Conv, Gate, Out) | 511M | 1.7B | 9.28 | 8.92 | 8.85 |
| + RoM (Conv, Gate, Out) w/ Bal. Loss | 511M | 1.7B | 9.30 | 8.94 | 8.87 |
| + RoM (Conv, Gate, dt, x, Out) | 511M | 1.7B | 9.30 | 8.93 | 8.87 |
| + RoM (Conv, Gate, dt, x, Out) w/ Bal. Loss | 511M | 1.7B | 9.26 | 8.89 | 8.83 |

In this section, we conduct experiments to evaluate whether introducing a load balance loss for regulating the global expert load distribution is necessary. Specifically, we adopt the widely used load balancing loss to promote a more even distribution of computational load across experts, aiming

to mitigate potential imbalances and enhance efficiency, formulated as:

$$\mathcal{L}_{bal} = \alpha \cdot \sum_{j=1}^{M} N \cdot \sum_{i=1}^{N} F_{i,j} \cdot \mathbb{E}(\text{Softmax}(\mathbf{X}_{i,j} \cdot \mathbf{W}_{r,j})) \tag{16}$$

where $\alpha$ is a hyperparameter and set to 1e-3 during training, $N$ is the number of experts, $M$ is the number of model layers, $F_{i,j}$ is the fraction of tokens dispatched to expert $i$ in layer $j$, and $\mathbf{W}_{r,j}$ is a set of trainable weights of the router in layer $j$.

The results in Table 6 compare the validation context length across different model configurations, with and without the load balance loss. The results show that adding load balance loss does not significantly improve performance and may slightly reduce it. The Samba (expand=4) baseline and RoM variants exhibit minimal fluctuations in validation context length, indicating that expert activation could be naturally balanced. The RoM (Conv, Gate, dt, dx, Out) configuration shows similar behavior, reinforcing that additional balancing constraints do not yield noticeable benefits. The small variances observed (e.g., 9.30 vs. 9.26 for 4K length) indicate that the system's built-in routing and expert selection mechanism effectively distributes computational load, making the explicit balancing loss redundant. This suggests that explicit load balance loss regularization could be unnecessary in RoM, as the system's built-in routing effectively distributes the computational load. The findings confirm that our approach ensures efficient training and reduces communication overhead without compromising scalability.

**More router details.** We adopt the standard practice in conventional MoE training to add jitter noise to expert routing, which would lead to implicit expert sampling [25]. Meanwhile, we also adopt SparseMixer [28, 29], which can estimate the gradient related to expert routing for better convergence.

### A.4 More Experimental Results

Table 7: Detailed results of performance for Mamba and RoM architectures across different configuration in Figure 4(a).

| Architecture | Active Params. | Total Params. | Validation Context Length | | | |
|---|---|---|---|---|---|---|
| | | | 4096 | 8192 | 12288 | 16384 |
| Mamba | 115M | 115M | 14.31 | 13.89 | 13.71 | 13.70 |
| RoM | 115M | 710M | 12.33 | 11.97 | 11.82 | 11.79 |
| Mamba | 353M | 353M | 11.18 | 10.83 | 10.68 | 10.65 |
| RoM | 353M | 2.5B | 10.00 | 9.69 | 9.55 | 9.52 |
| Mamba | 765M | 765M | 9.84 | 9.52 | 9.37 | 9.34 |
| RoM | 765M | 5.5B | 8.95 | 8.68 | 8.55 | 8.52 |
| Mamba | 1.3B | 1.3B | 9.07 | 8.76 | 8.63 | 8.60 |
| RoM | 1.3B | 10B | 8.52 | 8.27 | 8.15 | 8.12 |

### A.5 More experimental Results of Hybrid RoM + FFN-MoE

**Efficient Scaling with Hybrid RoM + FFN-MoE.** A key insight from Table 10 is that RoM effectively enhances hybrid MoE architectures by integrating expert selection across different layers. When comparing Samba + RoM + FFN-MoE (8top1) with Samba + FFN-MoE (16top1) in the 20B token setup, RoM achieves similar perplexity across all validation context lengths, with only a marginal increase in total parameters (2.9B vs. 2.8B). Similarly, Samba + RoM + FFN-MoE (16top1) significantly outperforms Samba + FFN-MoE (32top1) across all context lengths, achieving an 8.98 perplexity score at 4K length, compared to 8.88 for the FFN-MoE baseline. This similarly happens at 16K length, where RoM achieves 8.54 compared to 8.45, showing its ability to match the FFN-MoE (16top1) baseline with fewer parameters (5.6B vs. 5.7B).

Table 8: Detailed results of performance for Mamba and RoM architectures across different configuration in Figure 4(b).

| Architecture | Active Params. | Total Params. | Validation Context Length | | | |
|---|---|---|---|---|---|---|
| | | | 4096 | 8192 | 12288 | 16384 |
| Mamba | 115M | 115M | 14.42 | 14.00 | 13.82 | 13.80 |
| RoM | 115M | 710M | 12.48 | 12.10 | 11.93 | 11.90 |
| Mamba | 353M | 353M | 11.29 | 10.91 | 10.75 | 10.72 |
| RoM | 353M | 2.5B | 10.07 | 9.74 | 9.60 | 9.56 |
| Mamba | 765M | 765M | 9.95 | 9.63 | 9.48 | 9.45 |
| RoM | 765M | 5.5B | 9.04 | 8.75 | 8.61 | 8.57 |
| Mamba | 1.3B | 1.3B | 9.16 | 8.86 | 8.71 | 8.67 |
| RoM | 1.3B | 10B | 8.52 | 8.25 | 8.12 | 8.08 |

Table 9: Detailed results of performance for Mamba and RoM architectures across different configuration in Figure 4(c).

| Architecture | Active Params. | Total Params. | Validation Context Length | | | |
|---|---|---|---|---|---|---|
| | | | 4096 | 8192 | 12288 | 16384 |
| Mamba | 115M | 115M | 14.59 | 14.14 | 13.96 | 13.93 |
| RoM | 115M | 710M | 12.77 | 12.37 | 12.20 | 12.16 |
| Mamba | 353M | 353M | 11.46 | 11.06 | 10.90 | 10.86 |
| RoM | 353M | 2.5B | 10.14 | 9.81 | 9.65 | 9.61 |
| Mamba | 765M | 765M | 10.00 | 9.65 | 9.49 | 9.45 |
| RoM | 765M | 5.5B | 9.11 | 8.81 | 8.66 | 8.63 |
| Mamba | 1.3B | 1.3B | 9.19 | 8.87 | 8.72 | 8.68 |
| RoM | 1.3B | 10B | 8.65 | 8.38 | 8.23 | 8.19 |

Table 10: Comparison of model performance for FFN-MoE and hybrid RoM + MoE architectures across different sparsity level of experts.

| Architecture | Active Params. | Total Params. | Validation Context Length | | |
|---|---|---|---|---|---|
| | | | 4096 | 8192 | 16384 |
| **20B training tokens with micro batch 4 on 8×A100 GPUs** | | | | | |
| Samba + FFN-MoE (16top1) | 511M | 2.8B | 8.80 | 8.46 | 8.40 |
| Samba + RoM + FFN-MoE (8top1) | 511M | 2.9B | 8.83 | 8.49 | 8.43 |
| **17B training tokens with micro batch 1 on 16×A100 GPUs** | | | | | |
| Samba + FFN-MoE (32top1) | 511M | 5.7B | 8.88 | 8.60 | 8.45 |
| Samba + RoM + FFN-MoE (16top1) | 511M | 5.6B | 8.98 | 8.70 | 8.54 |

The ability to integrate MoE beyond FFN layers introduces a new paradigm in sparse mixture of Mamba linear projection experts, where hybrid RoM + FFN-MoE leverages the unique strengths of both state space models and FFN layers. Rather than relying solely on FFN-MoE to scale model capacity, hybrid RoM + FFN-MoE utilizes Routing Mamba to achieve expert selection, optimizing model efficiency without compromising quality. This approach suggests that hybrid RoM + FFN-MoE architectures could become a new gold standard, offering an innovative, scalable, and cost-effective alternative to traditional FFN-MoE approaches.

Table 11: Comparison of training throughput under different configurations.

| Architecture | Active Params. | Total Params. | Training Speed ($\times 10^5$ tokens/s) |
|---|---|---|---|
| **20B training tokens on 8$\times$A100 GPUs** | | | |
| Samba (expand=2) | 421M | 421M | 4.42 |
| + RoM (Conv, Gate, Out) | 421M | 1.0B | 3.57 |
| Samba (expand=4) | 511M | 511M | 3.34 |

## A.6 Training Throughput Results of RoM

We compare RoM models of two different sizes to their corresponding dense models with the same number of parameters, measuring their training throughput using the identical hardware. A key insight is that RoM outperforms simple layer width expansion.

