# OpenReview forum: "Routing Mamba: Scaling State Space Models with Mixture-of-Experts Projection"
_NeurIPS.cc/2025/Conference — NeurIPS 2025 poster_

### Official Review · Reviewer_VBDT · 2025-07-02

**Clarity:** 2
**Significance:** 2
**Originality:** 3
**Rating:** 4
**Confidence:** 3

**Summary:**

The authors propose Routing Mamba (RoM) to address the challenge of scaling the expressive power of Linear State Space Models (LSSMs) for long-sequence modeling. While LSSMs are gaining attention as efficient alternatives to Transformers, the naïve application of Mixture-of-Experts (MoE) often results in performance degradation or training failure. RoM mitigates this by sharing routing decisions across the projection layer and lightweight submodules within each Mamba block, leveraging synergy among experts. The model matches the performance of a much larger dense Mamba with only 1.3B active parameters—2.3× fewer—and achieves a 23% reduction in FLOPs.

**Questions:**

- The motivation behind several design choices in RoM is unclear. In Section 4.1, the paper shows that naïvely applying MoE to Mamba leads to degraded performance, and argues that simply attaching MoE modules independently to each unit is insufficient. However, in Section 4.2, the explanation jumps directly to the proposed solution without a clear analysis of why the naïve approach fails. Phrases like “focusing on the most impactful layers” or “lowering router learning difficulty” are used, but remain abstract without further justification. It would strengthen the paper if the rationale behind these claims were clarified and better supported.

- In particular, the claim that reusing routing decisions helps reduce the difficulty of router learning could be made more convincing. A comparison between independent and shared routers—for example, showing PPL vs. training steps or identifying early plateau points—would help support this argument empirically.

**Ethical Concerns:**

["NO or VERY MINOR ethics concerns only"]

**Final Justification:**

The rebuttal provided a clear explanation of the “functional fragmentation” issue in naïve MoE designs and how shared routing restores functional coherence in Mamba blocks. Empirical evidence (PPL vs. training steps) convincingly supports the claim that shared routing improves efficiency and stability over independent MoE. These points resolved my earlier technical concerns regarding the motivation and mechanism of the proposed approach.

However, the broader positioning of the work’s contributions beyond the LSSM/MoE context remains less clear, which limits its perceived generality and significance. Given the strong technical soundness but narrower perceived scope, I am keeping my original score unchanged.

**Limitations:**

yes

**Paper Formatting Concerns:**

No significant formatting issues were found.

**Quality:**

3

**Strengths And Weaknesses:**

Strengths
The paper clearly demonstrates that applying MoE naïvely to Mamba can degrade performance and proposes a concrete method to resolve this. Through comprehensive experiments, the authors show that RoM consistently improves performance across various context lengths. It also outperforms alternative strategies like MoA, Switch Transformer heads, and even naive Mamba layer expansion.

Weaknesses
Although the proposed method is empirically effective, the paper somewaht lacks a clear explanation of why the naïve MoE approach fails and why the proposed design works. The motivation behind the architectural choices is somewhat opaque, and statements like "focusing on “he most impactful layers” or "reducing router learning difficulty” remain abstract. A deeper discussion of the underlying failure modes and the rationale behind each design decision would help readers better understand and generalize the approach.

---

> ### Author Rebuttal · Authors · 2025-07-31
>
> We thank Reviewer VBDT for acknowledging the clarity of our problem statement and the strength of our experimental results.
> ***
>
> **Weakness and Q1: The motivation behind several design choices in RoM is unclear... why the naïve approach fails.**
>
> The naive MoE approach fails due to what we term functional fragmentation. In Mamba, the Conv, Gate, and Output projection layers are not independent modules; they are **functionally interdependent** and work in concert. A naive MoE with separate routers for each layer shatters this synergy. For any given token, its representation might be sent to `Conv Expert A`, `Gate Expert B`, and `Output Expert C`. Since these experts are trained independently, they cannot learn to coordinate, resulting in an incoherent and fragmented processing pathway. This creates an exceptionally difficult learning problem for the routers, which must somehow learn three independent, yet functionally-aligned, routing policies.
>
> The empirical consequence of this fragmentation is not just suboptimal performance, but a near-complete failure to learn effectively, as demonstrated in **Figure 2** and the training dynamics table in our response to **Q2**. The `Independent MoE` model's perplexity stagnates and remains worse than the dense baseline, providing clear evidence of this difficult, fragmented learning problem.
>
> RoM's shared routing directly solves this. By using a single router to select a unified `expert pathway` (e.g., `Conv Expert A`, `Gate Expert A`, `Output Expert A`), we restore the functional coherence of the Mamba block. This drastically simplifies the router's learning task from coordinating three independent policies to learning a single, unified one. This allows the expert pathways to co-adapt and learn specialized functions synergistically, which is why RoM learns efficiently and achieves superior performance. We have further expanded our intuition and discussion in **Section 5.3, Appendix A.7 and A.8** and will ensure it is highlighted more clearly in the main text.
> ***
>
> **Q2: The claim that reusing routing decisions helps reduce the difficulty of router learning could be made more convincing... A comparison between independent and shared routers... would help**.
>
> We thank the reviewer for pointing out how we can provide more direct evidence. Based on our training results, we have compiled the PPL at various training steps for the `Samba 421M` model. The comparison is between the `MoE-Mamba (Conv, Gate, Out)` strategy (`Independent MoE`) and our `RoM (Conv, Gate, Out)` strategy (`Shared Router`).
>
> | Training Steps | Independent MoE (PPL) $\downarrow$ | RoM-Shared Router (PPL) $\downarrow$|
> |:----------------:|:------------------------:|:---------------------------:|
> | 1000          | 16.84                  | **15.58**                 |
> | 2000          | 13.73                  | **12.91**                 |
> | 3000          | 12.50                  | **11.78**                 |
> | 4000          | 11.78                  | **11.12**                 |
> | 5000          | 11.28                  | **10.65**                 |
> | 6000          | 10.89                  | **10.30**                 |
> | 7000          | 10.60                  | **10.04**                 |
> | 8000          | 10.40                  | **9.91**                  |
> | 9000          | 10.26                  | **9.77**                  |
>
> As the table clearly shows, while the Independent MoE's perplexity does improve over time, its learning is significantly slower and less effective than our RoM. At every stage of training, RoM achieves a substantially lower perplexity, demonstrating its superior learning efficiency. The final 9000-step evaluation results for both models are also reported in **Table 1, Figure 2 and Table 4 in Appendix A.1**. At the end of training, RoM reaches a PPL of **9.77**, whereas the Independent MoE only manages **10.26**, a result that is still worse than the final dense baseline (10.05 PPL).
>
> This large and consistent performance gap provides direct empirical support for our claim that the shared routing strategy reduces the difficulty of router learning, allowing the model to effectively and efficiently leverage the expert capacity from the very beginning of training. We will add this table and analysis to the appendix and refer to it in the main text. We also commit to adding a PPL vs. training steps plot in the final camera-ready version to visualize this compelling trend.

---

> > ### Comment · Reviewer_VBDT · 2025-08-09
> >
> > Thank you for the detailed and well-structured rebuttal. Your explanation of the “functional fragmentation” issue in the naïve MoE design, and how it disrupts the synergy between the Conv, Gate, and Output projection layers in Mamba, was clear. The unified routing rationale—restoring functional coherence and simplifying the router’s learning objective—makes the design motivation much more concrete. I also found the empirical evidence in the PPL vs. training steps comparison convincing in showing the efficiency and stability gains of RoM over the independent MoE approach. This resolves much of the ambiguity I initially noted regarding why the naïve approach fails and how the proposed design addresses it.
> >
> > That said, while the paper is technically sound and empirically rigorous, I believe there is room to further enhance its perceived significance. The work is highly relevant for the LSSM and MoE communities, and it could be even more impactful if its underlying principles were positioned in a way that highlights potential applicability to other architectures or theoretical contexts. This is not strictly necessary for the contribution to be valid, but doing so may broaden its appeal and influence. For this reason, I will keep my score unchanged, even though my earlier technical doubts have been largely resolved.

---

> ### Author Response · Authors · 2025-08-05
>
> Dear reviewer VBDT,
>
> As we approach the end of the discussion period, we would greatly appreciate your input on the paper. We hope our responses and additional results address your concerns and welcome any further questions or suggestions.
>
> Thank you for your time and effort.

---

### Official Review · Reviewer_KHjA · 2025-07-03

**Clarity:** 3
**Significance:** 3
**Originality:** 3
**Rating:** 4
**Confidence:** 3

**Summary:**

This paper presents a careful study on applying Mixture-of-Experts techniques to State Space Models (SSMs). It demonstrates that naively routing different parts of the model independently does not yield strong performance. Instead, sharing routing strategies via carefully chosen projection matrices results in better outcomes. The proposed method, RoM, shows promise across multiple configurations and tasks.

**Questions:**

See `Weaknesses`

**Ethical Concerns:**

["NO or VERY MINOR ethics concerns only"]

**Limitations:**

See `Weaknesses`

**Quality:**

3

**Strengths And Weaknesses:**

**Strengths**

1. The paper conducts a thorough and well-scoped investigation into where and how sparse structures can be applied within SSMs. It clearly demonstrates that straightforward sparsification is not sufficient.

2. The authors identify effective strategies for layer selection and routing design, leading to a generalizable approach that applies across multiple model sizes and context lengths. On smaller model scales, RoM achieves significant gains over larger dense baselines.

3. The paper includes thoughtful design choices—such as eliminating the need for a load-balancing loss—while still achieving stable training, which highlights the practicality and robustness of the method.

**Weaknesses**

The paper does not extensively analyze the trade-offs between routing overhead and performance gains, especially as the number of experts increases, or selecting at least 2 experts per-token. A deeper cost-benefit discussion would strengthen the practical relevance.

---

> ### Author Rebuttal · Authors · 2025-07-31
>
> We thank Reviewer KHjA for recognizing our thorough investigation and thoughtful design choices.
> ***
>
> **Weaknesses: The paper does not extensively analyze the trade-offs... as the number of experts increases or selecting at least 2 experts per-token**.
>
> This is an excellent point. To directly address the reviewer's question, we have conducted additional experiments to analyze these trade-offs. The following table presents the measured training throughput for various RoM configurations, extending the setting from **Table 11 in Appendix A.6**.
>
> | Configuration | Total Experts | Activated Experts | Throughput (tokens/s) $\uparrow$ |
> | :------------ | :------------:| :----------------:| :---------------------:|
> | 8top1     | 8             | 1                 | $3.57\times10^5$            |
> | 16top1    | 16            | 1                 | $3.10\times10^5$            |
> | 8top2     | 8             | 2                 | $2.65\times10^5$            |
> | 16top2    | 16            | 2                 | $2.27\times10^5$            |
>
>
> **Analysis of Trade-offs**:
>
> 1. **Increasing Total Experts (e.g., 8 to 16)**: Doubling the total number of experts while keeping `k=1` (8top1 vs. 16top1) results in a ~13% throughput reduction. This demonstrates that increasing the model's total parameter capacity incurs a manageable overhead.
> 2. **Increasing Activated Experts (e.g., top-1 to top-2)**: Doubling the number of activated experts for a fixed set of 8 total experts (8top1 vs. 8top2) results in a ~26% throughput reduction.
>
> These results provide a clear quantitative picture of the trade-offs involved. Increasing the total number of experts is a relatively cheap way to scale model capacity, while activating more experts is more costly but may yield higher performance. We thank the reviewer for this valuable suggestion; we will add this table and analysis to the experimental results section in the final manuscript to provide a more complete picture of RoM's scaling properties.

---

> ### Author Response · Authors · 2025-08-05
>
> Dear reviewer KHjA,
>
> As we approach the end of the discussion period, we would greatly appreciate your input on the paper. We hope our responses and additional results address your concerns and welcome any further questions or suggestions.
>
> Thank you for your time and effort.

---

### Official Review · Reviewer_bjjv · 2025-07-03

**Clarity:** 3
**Significance:** 3
**Originality:** 3
**Rating:** 4
**Confidence:** 2

**Summary:**

This paper proposes a new method to scale State Space Models (SSMs), like Mamba, with Mixture-of-Experts. The authors indicate that naive integration of MoE into Mamba's architecture is ineffective and can degrade performance. The primary contribution is a novel method called Routing Mamba (RoM), which applies a sparse mixture of experts to the linear projection layers of Mamba. Through extensive experiments, the paper demonstrates that RoM outperforms standard dense Mamba models..

**Questions:**

N/A

**Ethical Concerns:**

["NO or VERY MINOR ethics concerns only"]

**Final Justification:**

While this work is a good execution, its significance is limited, so I am keeping my initial rating.

**Limitations:**

Yes

**Quality:**

3

**Strengths And Weaknesses:**

Strengths:
- The authors conducted a comprehensive set of experiments. The authors validate their approach across multiple model scales, demonstrating consistent and significant performance gains.
- The paper is well-written, clear, and easy to follow.

Weaknesses:
- The contribution, while effective, could be viewed as incremental. The central novel idea is the shared routing strategy, which could be seen as a well-executed "engineering trick" rather than a novel algorithm since it lacks theoretical results to support the method. The explanation is a bit heuristic. There is no formal analysis to prove why this approach should lead to more stable training or better expert specialization.

---

> ### Author Rebuttal · Authors · 2025-07-31
>
> We thank Reviewer bjjv for the positive feedback on the clarity of our paper and the comprehensive nature of our experiments.
> ***
>
> **Weakness: The contribution, while effective, could be viewed as incremental... an "engineering trick"**.
>
> We respectfully argue that our contribution is more fundamental than an "engineering trick." The core challenge we address is that **naive MoE integration in Mamba fails**, a fact we demonstrate empirically in **Figure 2** and which is consistent with prior work. Simply applying MoE to Mamba's projection layers independently, arguably the real straightforward "engineering" approach (`Independent MoE`), which not only fails to improve upon the dense baseline but actively degrades performance (**10.26** PPL  vs. **10.05** PPL).
>
> Our primary contribution is identifying this critical failure mode and proposing a novel architectural solution: a shared routing strategy that creates a coherent `expert pathway`. This design is non-trivial and is the key that unlocks, for the first time, the effective scaling of Mamba-style SSMs with MoE. It is a targeted solution to a documented instability. The resulting **2.3 $\times$ gain in active parameter efficiency** (**Figure 3**) underscores the significance as a novel and effective architectural approach, not merely an incremental tweak.
> ***
>
> **Lack of theoretical results to support the method... explanation is a bit heuristic... no formal analysis to prove... more stable training or better expert specialization.**.
>
> We thank the reviewer for this insightful comment and agree that our paper's primary contributions are empirical. While we do not provide a formal theoretical proof, our explanation is grounded in a testable hypothesis that our experiments were designed to validate.
>
> Our central hypothesis is that the naive MoE approach fails due to functional fragmentation. The Mamba block's projection layers (Conv, Gate, Output) are functionally interdependent. Using independent routers forces the model to learn three separate, uncoordinated routing policies, creating an exceptionally difficult optimization problem. Our results provide direct evidence for this failure. As shown in our response to Reviewer VBDT (Q2), the `Independent MoE` model exhibits an unstable learning curve and its final perplexity of **10.26** is worse than the dense baseline (10.05). This outcome is a direct validation of our hypothesis: the independent routers fail to converge to an effective policy.
>
> In contrast, RoM's shared routing mechanism is designed specifically to enforce functional coherence. By using a single router, we simplify the learning problem to finding one unified policy, allowing the expert pathways to co-adapt. The empirical validation is clear: the RoM model learns stably, consistently outperforms the naive approach at every checkpoint, and achieves a superior final PPL of **9.77**.
>
> Therefore, the clearly different outcomes of these two controlled experimental conditions provide strong, causal evidence for our explanation, moving it beyond a mere heuristic to an empirically-verified principle for scaling SSMs.
>
> **We would like to highlight that our results provide direct empirical evidence that validates our approach leads to both more stable training and better expert specialization**.
>
> **Evidence for More Stable Training**:
> Our experiments provide two key pieces of evidence for training stability.
>
> 1. **Consistent Learning Dynamics**: As shown in the PPL vs. training steps table (in our response to Reviewer VBDT), the naive `Independent MoE` model exhibits unstable learning, with its final perplexity (10.26) being worse than the dense baseline (10.05). In contrast, our RoM model shows a smooth, consistent decrease in perplexity at every configuration, demonstrating a stable and effective learning process.
>
> 2. **Inherent Load Balancing**: As highlighted in **Section 4.3 and Appendix A.3 (Table 6)**, RoM trains effectively without **requiring an auxiliary load-balancing loss**, a common mechanism used to stabilize MoE training. This inherent stability is a direct result of our shared routing design.
>
> **Evidence for Better Expert Specialization**:
> The superior performance of RoM is a direct consequence of more effective expert specialization.
>
> 1. **Superior Final Performance**: The significant gap in final perplexity between RoM (**9.77**) and the naive approach (**10.26**) strongly indicates that RoM's experts have learned more specialized and effective functions. If the specialization were not better, the performance would not be superior.
>
> 2. **Efficient Scaling**: The consistent scaling results in **Figure 3**, where RoM achieves performance equivalent to a **2.3**$\times$ larger dense model, would not be possible without effective specialization. Poor specialization would lead to diminishing returns, but RoM demonstrates a clear and efficient scaling advantage, which is a hallmark of successful expert specialization.
>
> Therefore, while the initial intuition may seem a bit heuristic, it is directly and rigorously tested. We have empirically identified a specific failure mode in naive MoE for SSMs and demonstrated that our proposed architectural principle, shared routing for functional coherence, is the solution. We believe this data-driven approach provides a solid, non-heuristic foundation for our claims, with formal analysis being a promising avenue for future work.

---

> ### Author Response · Authors · 2025-08-05
>
> Dear reviewer bjjv,
>
> As we approach the end of the discussion period, we would greatly appreciate your input on the paper. We hope our responses and additional results address your concerns and welcome any further questions or suggestions.
>
> Thank you for your time and effort.

---

> > ### Comment · Reviewer_bjjv · 2025-08-08
> >
> > Thank you for the rebuttal, and it addresses most of my concern, however, I still think this work is limited significance, and I maintain my score.

---

### Official Review · Reviewer_ceRt · 2025-07-05

**Clarity:** 4
**Significance:** 3
**Originality:** 4
**Rating:** 5
**Confidence:** 3

**Summary:**

The paper introduces Routing Mamba (RoM), a novel framework that integrates Mixture-of-Experts (MoE) into State Space Models (SSMs), specifically Mamba, to enhance scalability and efficiency in sequence modeling.

Naive MoE integration into Mamba layers degrades performance due to the complex interplay of projection layers (Conv, Gate, Output).

Therefore, RoM solves this issue by employing a shared routing strategy across the projection layers, selectively scaling impactful components while reusing routing decisions to reduce overhead and improve coherence. In RoM, only the top $K$ experts corresponding to the highest gating weigths $\\mathcal{R}_i(\\mathbf{X}_t)$ are selected. Where the gating is performed through a routing projection layer $\\mathbf{W}_r$, followed by a SoftMax for obtaining probabilities.

The method is evaluated on Mamba with parameters varying from 115M to 1.3B, where 1 out of 8 expert networks for each token at each layer is activated. The models are trained on 20B tokens of SlimPajama, and evaluated on its validation set. Additionally, the approach is tested on several other zero-shot datasets: LAMBADA, HellaSwag, PIQA, ARC-Easy, ARC-Challenge, and WinoGrande. The results show clear dominance of RoM over the baseline Mamba, under the same number of active parameters.

**Questions:**

N/A

**Ethical Concerns:**

["NO or VERY MINOR ethics concerns only"]

**Final Justification:**

After reading the other reviews and the authors' response, I have decided to maintain my current score.

**Limitations:**

Yes

**Paper Formatting Concerns:**

- In the abstract the paper mentions that SSMs have "constant inference-time computation and memory complexity", this is obviously not true and needs to be fixed.
- The definition of the SSM (lines 124 to 127) is not good, the notations have $L$ in them, which is the sequence length defined in line 116. Please reformulate this definition.

**Quality:**

3

**Strengths And Weaknesses:**

**Strengths:**

- The paper is well written.
- The approach is properly tested on multiple datasets.
- The contribution of each block is studied.
- RoM matches the performance of dense Mamba models with 2.3× fewer active parameters.
- Achieves 23% FLOPs reduction in hybrid SSM-attention models (e.g., Samba).
- Maintains consistent perplexity across varying context lengths (4K–16K).

**Weaknesses:**
- While RoM reduces active parameter usage and FLOPs, training throughput still suffers.
- Although shared, the routing mechanism introduces new design challenges and could affect training stability in less controlled settings.

---

> ### Author Rebuttal · Authors · 2025-07-31
>
> We thank Reviewer ceRt for the positive assessment and for recognizing the novelty and significance of our results.
> ***
>
> **W1: Training throughput still suffers**.
>
> We thank the reviewer for this crucial point on training efficiency. While RoM with our current FSDP-only setup has a lower raw throughput than a dense model of the same active size, we argue this represents a **highly favorable trade-off for achieving superior model quality**.
>
> Our choice to use a pure FSDP setup, as detailed in Section 5.1, was a **deliberate methodological decision**. This approach, while not the fastest for MoE models, ensures a **fair and stable comparison** by eliminating confounding variables like expert parallelism (EP), capacity factors, and token dropping. This allows us to isolate and measure the architectural benefits of RoM itself.
>
> More importantly, when comparing RoM to a dense model of **comparable performance**, RoM is actually **more computationally efficient**. Based on **Table 11 in Appendix A.6**:
>
> 1. Our `RoM (Conv, Gate, Out)` model (421M active params) has a training throughput of $3.57 \times 10^5$ tokens/s.
> 2. To achieve a comparable performance, the dense model needs to be scaled up to `Samba (expand=4)` (511M active params), which has a training throughput of only $3.34\times10^5$ tokens/s.
>
> This comparison clearly shows that our RoM model not only achieves better performance with fewer active parameters but is also **faster to train than a comparable, larger dense model**. Therefore, RoM offers a more efficient path to higher performance than simply scaling up a dense model.
>
> Furthermore, the throughput of RoM can be **significantly accelerated** by integrating standard optimizations like **expert parallelism** or **better CUDA/Triton kernels designed for MoE**, which we leave for future engineering work. Our current results already demonstrate a compelling efficiency frontier, and with further optimization, the practical benefits of RoM will be even greater.
> ***
>
> **W2: Shared routing introduces new design challenges and could affect training stability**.
>
> This is a valid consideration. We are pleased to report that our experiments demonstrate remarkable training stability.
>
> 1. **Consistent Performance**: As shown in **Figure 3**, RoM delivers consistent and significant perplexity improvements across all tested model scales (115M/710M to 1.3B/10B) and context lengths (4K to 16K), indicating that the shared routing mechanism is stable and generalizes well.
> 2. **Inherent Load Balancing**: A key finding, highlighted in **Section 4.3** and **Appendix A.3** (**Table 6**), is that RoM **does not require an auxiliary load-balancing loss for small scale**. The routing mechanism naturally distributes the load across experts, which points to the inherent stability and robustness of our shared routing design. This simplifies the training process compared to many traditional MoE models.
> ***
>
> **Formatting Concern 1: "constant inference-time computation" in the abstract**.
>
> Thank you for this precise correction. You are right that the complexity is dependent on the sequence length. We will revise this statement in the abstract and introduction to be more accurate, clarifying that SSMs feature **linear-time computation complexity with respect to sequence length** and **constant memory complexity during the per-step recurrence**, which is the key advantage over the quadratic complexity of standard Transformers.
>
> **Formatting Concern 2: The definition of the SSM (lines 124-127)**.
>
> We appreciate you spotting this notational misuse. We will revise this part and ensure the definitions are rigorous and clear.

---

> ### Author Response · Authors · 2025-08-05
>
> Dear reviewer ceRt,
>
> As we approach the end of the discussion period, we would greatly appreciate your input on the paper. We hope our responses and additional results address your concerns and welcome any further questions or suggestions.
>
> Thank you for your time and effort.

---

### Note · Authors · 2025-08-13

We sincerely thank all reviewers and the Area Chair for their time and valuable feedback. We are encouraged that the reviewers found our work "well written" (ceRt, bjjv), "thorough" (KHjA), and supported by "comprehensive experiments" (bjjv, VBDT). We appreciate the constructive comments, which will help us improve the final version of our paper. We are also grateful for the all positive scores from the entire panel.

We are pleased that our rebuttals and additional experiments have addressed most of the reviewers' technical concerns, particularly regarding the failure of naive MoE integration and the stability and effectiveness of our proposed RoM framework. As Reviewer VBDT noted, our explanation of "functional fragmentation" and the empirical evidence provided have resolved the initial ambiguity.

Our key contribution is a novel and non-trivial architectural solution to a documented failure mode in scaling Mamba-style SSMs with MoE. The resulting 2.3x gain in active parameter efficiency and the 23% FLOPs reduction in hybrid models are significant empirical results that move beyond a simple "trick".

We believe RoM offers a new and effective principle for scaling SSMs, and we are confident that this work will inspire future research into more efficient and powerful sequence models. We have incorporated the feedback to improve the clarity and rigor of the paper and look forward to sharing our work with the broader community.

---

### Decision · Program_Chairs · 2025-09-17

**Decision:**

Accept (poster)

**Comment:**

This paper introduces Routing Mamba (RoM), a novel approach to scale State Space Model parameters using sparse mixtures of linear project experts. The authors demonstrate the effectiveness of the proposed approach through detailed experiments conducted on 1.3 billion active parameters, using a sequence length of 16,000. RoM got the same performance with a dense Mamba with 2.3x more active parameters and saved 23% FLOPs.

I agree with the feedback from all reviewers. The authors addressed questions and concerns in detail in their rebuttal. In further work, I'd like to encourage the authors to conduct distributed learning experiments to enhance the practical value of model training.